# Clinical Impacts of Interventions for Physical Activity and Sedentary Behavior on Patients with Chronic Obstructive Pulmonary Disease

**DOI:** 10.3390/jcm12041631

**Published:** 2023-02-17

**Authors:** Hiroki Tashiro, Koichiro Takahashi

**Affiliations:** Division of Hematology, Respiratory Medicine and Oncology, Department of Internal Medicine, Faculty of Medicine, Saga University, Saga 849-8501, Japan

**Keywords:** chronic obstructive pulmonary disease, physical activity, sedentary behavior

## Abstract

Recently, physical activity has increasingly become the focus in patients with chronic obstructive airway disease (COPD) because it is a strong predictor of COPD-related mortality. In addition, sedentary behavior, which is included as a category of physical inactivity including such behaviors as sitting or lying down, has an independent clinical impact on COPD patients. The present review examines clinical data related to physical activity, focusing on the definition, associated factors, beneficial effects, and biological mechanisms in patients with COPD and with respect to human health regardless of COPD. The data related to how sedentary behavior is associated with human health and COPD outcomes are also examined. Lastly, possible interventions to improve physical activity or sedentary behavior, such as bronchodilators and pulmonary rehabilitation with behavior modification, to ameliorate the pathophysiology of COPD patients are described. A better understanding of the clinical impact of physical activity or sedentary behavior may lead to the planning of a future intervention study to establish high-level evidence.

## 1. Introduction

Chronic obstructive pulmonary disease (COPD) is a common respiratory disease, with 251 million cases worldwide, and it is a life-threatening disease that became the third leading cause of death globally in 2019 [1]. Medical interventions focusing on physical activity in the stable phase are currently demanded for COPD patients according to the results of several studies, although avoiding events such as exacerbations, infections, and comorbidities is also important because they are involved in clinical outcomes, including mortality, of COPD patients [2]. In addition, the specific impact of sedentary behavior on COPD patients has come to be understood [3]. Importantly, the effects of physical inactivity and sedentary behavior on clinical outcomes of COPD patients are different, and many data have shown that sedentary behavior negatively affects the pathophysiology of COPD, independent of physical inactivity, as described below.

In this review, the clinical impacts of physical activity and sedentary behavior on patients with COPD are addressed. The interactions of physical activity, sedentary behavior, and human health regardless of COPD are also described to explore novel therapeutic approaches. Lastly, possible therapeutic interventions focusing on sedentary behavior for COPD patients, including pharmacological and nonpharmacological interventions, such as pulmonary rehabilitation including behavior modification, are shown. We believe that the present review can contribute practically to clinical practice and provide future direction for the appropriate design of a clinical trial specifically focusing on sedentary behavior in COPD patients.

## 2. Physical Activity and Human Health

### 2.1. What Is Physical Activity?

Physical activity is defined as any movement or exercise of the body produced by skeletal muscles that requires energy consumption [4]. To focus on the clinical impact of physical activity on human health, the intensity level should be considered with respect to individual physical performance and response. Physical activity involves organs and tissues producing motion such as the skeletal muscles, heart, lungs, and others, and it leads to energy metabolism via aerobic or anaerobic pathways, depending on the individual threshold [5]. Aerobic metabolism requires oxygen to meet the energy demand during movement and uses energy derived from adenosine triphosphate (ATP) synthesized in the mitochondria of skeletal muscle by the tricarboxylic acid cycle [6,7]. Movements inducing aerobic metabolism are normally considered light- to moderate-intensity activities that allow people to perform typically repeating sequences of movements without causing risk or harm to human health [8,9]. On the other hand, anaerobic metabolism induces glycolysis, which is the transformation of glucose to lactate via the pyruvate cascade and synthesis of ATP under limited amounts of oxygen [10,11]. Movements inducing anaerobic metabolism are high-intensity activities that reach in excess of 90% of the maximum heart rate, increasing the risk of physical disorders such as musculoskeletal injuries along with the higher load on the body [12,13]. Thus, the recommended intensity level of an activity and its duration in daily life should be considered according to its benefit, especially for elderly individuals.

For standardized evaluation of physical activity, several methods including questionnaires and devices are used [14]. Briefly, questionnaires involve individual reporting of patients’ own physical activity behaviors, along with job categories, such as the global physical activity questionnaire, international physical activity questionnaire, and the short physical performance battery [14,15]. Devices such as pedometers that count steps and accelerometers that measure truncal or limb movements are normally used [14]. Physical activity considering intensity and duration has increasingly come into focus for the maintenance of human health and the improvement of disease pathophysiology. The intensity of physical activity is calculated by metabolic equivalents (METs) and defined as follows: sedentary behavior, 1.0–1.5 METs; light intensity, less than 3.0 METs; moderate intensity, 3.0–6.0 METs; vigorous intensity, more than 6.0 METs [16]. Examples of physical activities calculated by METs are shown in Table 1.

The recommended intensity of physical activity differs depending on life stage, purpose, and demand. For example, it is strongly recommended that children and adolescents do at least an average of 60 min per day of moderate- to vigorous-intensity physical activity [18,19,20]. For adults aged 18–64 years, 150–300 min of moderate-intensity aerobic physical activity throughout the week and muscle-strengthening activity at moderate or greater intensity involving all major muscles groups on two or more days a week is strongly recommended [21,22]. For older adults aged 65 years and older, at least 150–300 min of moderate-intensity aerobic physical activity or 75–150 min of vigorous-intensity aerobic physical activity throughout the week is recommended [21,23]. Muscle-strengthening activities such as exercise training at moderate or greater intensity can also be performed on two or more days a week, along with the physical activity protocol mentioned above [4]. At all ages, physical activity should be undertaken as part of recreation, leisure such as play, games, sports, or planned exercise, transportation, work, or household chores in daily occupational, educational, home, and community settings [4], but the appropriate intensity level of the activity should be considered depending on the physical condition, including COPD and comorbidities. According to these recommendations, physical activity should be encouraged for maintenance of human health, although it is necessary to avoid excessive load on skeletal muscles that could cause harm.

### 2.2. Factors Associated with Physical Activity

Considering the clinical impact of physical activity on human health, it is necessary to understand which factors are related to physical activity in daily life. The factors associated with physical activity are varied, and include physical disability, mental health, lifestyle, and environmental situation. Physical function, including bone mass and muscle strength, tends to decrease with increasing age, which is related to decreased physical activity [23,24,25]. Specifically, sarcopenia, which is an age-related decrease of skeletal muscle volume plus low muscle strength, is associated with decreased physical performance [26]. Osteopenia/osteoporosis, which is characterized by low bone mass, also increases the risk of fractures due to bone fragility, leading to decreased physical activity [27,28]. The range of motion of the ankle joint, which is important for smooth forward movement during walking, is restricted by trauma, aging, and inflammatory disease of the joint [29,30] and, importantly, limited joint mobility decreases physical activity [30]. Mental health also contributes to physical inactivity [31]. Cross-sectional analyses involving 1536 Germans showed that individuals who do not perform physical exercise have a 3.15 times increased risk of developing moderate to severe depression [32]. Furthermore, positive mood, emotions [33,34], and even sleep quality are significantly correlated with physical activity level [35]. Lifestyles of individuals [36], including use of cell phones and computers, as well as playing video games, are also related to reductions in steps per day, and these data remind us of the necessity of ‘behavior modification’ to improve physical activity. Certainly, a person’s environmental situation is a strong factor related to physical activity. Accordingly, various factors affect physical activity, and interventions for improvement require a multilateral focus.

### 2.3. Beneficial Effects of Physical Activity on Human Health and the Mechanisms

Improvement of physical activity produces beneficial effects on human health and quality of life (QOL) [4]. Evidence has shown that regular physical activity such as mat Pilates improves physical function, including muscle strength, flexibility, and cardiorespiratory fitness [37]. Importantly, physical activity increases life expectancy [38]. A multicenter, cohort analysis in the US showed that a greater number of daily steps measured by accelerometer is significantly associated with lower all-cause mortality [39]. Notably, whether an intervention that increases physical activity would contribute to increasing life expectancy is not known because of the lack of clinical studies, but all-cause mortality might be reduced by improving wellbeing with increased individual physical activity. Furthermore, physical activity can contribute to health benefits for adults and older adults with chronic conditions, such as cancer survivors, or those with hypertension or type 2 diabetes mellitus. Briefly, a higher intensity of physical activity after cancer diagnosis has a protective effect on all-cause and cancer-specific mortality of various types of cancer [40]. For hypertension, people who engage in regular exercise show decreased systolic blood pressure of approximately 12 mmHg compared to people who do not exercise regularly [41]. Aerobic activity and/or muscle-strengthening activity are also related to improvements in glucose tolerance assessed as by glycosylated hemoglobin and insulin levels in patients with type 2 diabetes mellitus [42]. In terms of mental health, several studies have shown that engaging in physical activity and/or exercise programs can also improve emotional wellbeing [43]. Babyak et al. reported that an aerobic exercise intervention showed significantly better improvement and a lower relapse rate of depression than psychotropic treatment [44]. Similarly, a walking program in addition to social sports, measured using a 10-item modified version of the social support for exercise scale [45], was significantly associated with greater positive mood in women [46]. Notably, exercise, as mentioned above, is defined as planned, structured, and repetitive physical activity, not exactly identical to the broad meaning of physical activity [47]. However, exercise training has the capacity to improve physical activity with modification of behavior [47].

As for the biological mechanisms of the beneficial effects of improvement of physical activity, blunting or optimizing modulation of hormonal stress-responsive systems such as the hypothalamic–pituitary–adrenal axis and the sympathetic nervous system, which contribute to physiological, emotional, and metabolic reactivity, has been considered [48,49,50,51]. In addition, exercise affects the brain by enhancing growth factor expression and neural plasticity, contributing to improved mood and cognition [52,53]. A resilient anti-inflammatory effect through minimization of excessive inflammation is also induced by regular exercise/activity, which is supported by previous studies. Briefly, cancers, hypertension, and type 2 diabetes mellitus are associated with systemic markers of inflammation such as tumor necrosis factor-alpha, interleukin (IL)-1, IL-6, IL-8, and C-reactive protein, and exercise intervention reduces them [48]. As mentioned above, exercise intervention increases physical activity along with improvements of skeletal muscle function and cardiopulmonary function, and the improvement might attenuate systemic inflammation [54,55]. According to these data, exploring biological targets related to improvement of physical activity might be important for focusing on novel therapeutic perspectives.

## 3. Physical Activity and Chronic Obstructive Pulmonary Disease

### 3.1. Clinical Impact of Physical Activity in Patients with COPD

There is increasing evidence that decreased physical activity in patients with COPD is an important part of the pathophysiology that is associated with the clinical outcome. For example, Minakata et al. [56] reported that patients with COPD showed a significant reduction in the duration of physical activity, including activities at each intensity level of more than 2.0 METs, 2.5 METs, 3.0 METs, and 3.5 METs, compared to healthy subjects. In addition, levels of physical activity were further reduced with progression of the Global Initiative for Chronic Obstructive Lung Disease (GOLD) stage [56]. Similarly, walking and standing times are shorter, and time spent sitting and lying down is longer in patients with COPD than in healthy subjects [57]. Another single-center study showed that, in patients with COPD, daily physical activity measured by a triaxial accelerometer was an independent prognostic factor for mortality and hospitalization due to severe exacerbations [58]. Importantly, Waschki et al. [59] suggested, in a prospective, cohort study, that physical activity is the most important predictor of all-cause mortality. The relative risks of death with a standardized decrease in physical activity level measured by a multisensory armband and steps per day are higher than those with worsening of pulmonary function, exercise capacity assessed by 6 min walk distance, COPD-related QOL, or symptoms such as dyspnea in patients with COPD. 

Notably, sarcopenia [60], osteopenia [61], and depression [62] are negatively associated with physical activity, as mentioned above, and they are major comorbidities of COPD. Thus, a decreased level of physical activity in patients with COPD might be involved in these comorbidities. According to these data, physical activity has a great impact in patients with COPD, and aggressive intervention to increase physical activity might be necessary to improve clinical outcomes of patients with COPD. Therefore, the present GOLD guideline notes that physical activity is a strong predictor of mortality, and COPD patients might be encouraged to increase their physical activity levels [63]. Again, it is unknown whether interventions that increase physical activity in patients with COPD, such as pulmonary rehabilitation, would directly contribute to improvement in the clinical outcomes of COPD, such as mortality, QOL, symptoms, and exacerbations, because of a shortage of data. Recently, a standard for the recommended number of steps for patients with COPD considering the modified Medical Research Council (mMRC) dyspnea scale and inspiratory capacity (IC) has been reported. A simple standard equation is the following: step count = (−0.079 × [age] − 1.595 × [mMRC] + 2.078 × [IC] + 18.149)^3^ [64], which is a useful tool for education of patients with COPD.

### 3.2. Factors Associated with Physical Activity in Patients with COPD

There are several factors related to physical activity in patients with COPD. Pulmonary function parameters such as forced vital capacity (FVC) percent predicted and FEV1.0 percent predicted are positively correlated with physical activity [56]. Results on the incremental shuttle walk test and 6 min walking distance, muscle function measured by handgrip force and quadriceps force, and symptoms such as dyspnea are also associated with physical activity in patients with COPD [57,65]. Briefly, skeletal muscle is positively associated with exercise capacity as assessed by 6 min walking distance and oxygen uptake at peak exercise measured by cardiopulmonary exercise testing in patients with COPD [66,67]. Interestingly, the cross-sectional area of skeletal muscle measured by computed tomography is positively associated with physical activity [67], showing that exercise capacity involves physical activities via skeletal muscle mass. Waschiki et al. [65] reported that physical activity measured by a multisensory armband and steps per day is negatively correlated with score on the modified Medical Research Council dyspnea scale. These data show that pulmonary function, exercise tolerance involving skeletal muscles, and symptoms related to COPD, which are major pathophysiological markers and clinical features of COPD, affect physical activity. Yoshida et al. reported that depression and anxiety assessed using the self-rating depression scale and state-trait anxiety inventory were significantly correlated with physical activity level [68]. Others also found, in a prospective, multicenter study, that symptoms of depression examined by the Hospital Anxiety and Depression Scale [69] were associated with a measurable reduction in physical activity 6 months later in patients with COPD [70]. In addition to parameters related to the pathophysiology of COPD, sociodemographic factors including age, sex, cultural group, educational level, and working status might be involved in physical activity levels [71]. Lifestyle and environmental factors including alcohol consumption and smoking might also be associated with physical activity levels in patients with COPD [71].

### 3.3. Possible Biomarkers Reflecting Physical Activity of Patients with COPD

There is an increased focus on evaluation of biomarkers related to physical activity in patients with COPD. For example, the total cholesterol level in blood, which reflects cardiac function and nutritional status, 8-isoprostane in exhaled breath condensate (EBC), which is an airway oxidative stress marker, and IL-6 in EBC, which reflects systemic inflammation, are negatively correlated with physical activity level [68,72]. Myokines, especially irisin, which has been discovered as a hormone secreted from skeletal myocytes at the start of exercise training [73,74], is considered a valuable biomarker reflecting physical activity in COPD. Ijiri et al. [75] evaluated serum irisin levels, where were lower in patients with COPD than in control subjects. The irisin level is correlated with physical activity level, but it is not correlated with pulmonary function and 6 min walking distance in patients with COPD. Interestingly, 8 week exercise training is linked to a significant increase in the irisin level [75], suggesting that irisin might be a useful candidate biomarker reflecting physical activity in patients with COPD. Furthermore, decreased serum irisin levels are involved in epithelial apoptosis, resulting in emphysema in patients with COPD [76]. Others have reported that growth differentiation factor 11 (GDF-11) in plasma, which is expressed in skeletal muscle and is linked to rejuvenating effects such as muscle regeneration, was positively correlated with physical activity level. GDF-11 is increased with improvements of lung function, quadriceps strength, and exercise capacity, as well as reduced inflammatory markers in patients with COPD [77]. According to these data, irisin and GDF-11 are beneficial, and 8-isoprostane has a harmful effect for COPD patients with respect to physical activity.

### 3.4. Beneficial Effects of Physical Activity on Patients with COPD and the Mechanisms

Previous epidemiological data have shown that increased physical activity is associated with the possible improvement of all-cause mortality or COPD-related mortality [58,59]. However, to the best of our knowledge, effects on mortality by specific physical activity interventions have not been evaluated. A multidisciplinary, exercise-based program aimed to improve physical activity such as pulmonary rehabilitation is considered a beneficial intervention [78,79,80], and pulmonary rehabilitation is strongly recommended in the present COPD guideline [81], because functional disorders of skeletal muscle, such as frailty or sarcopenia, have harmful impacts on the pathophysiology of COPD, including on physical activity [82,83]. There are several lines of evidence showing that pulmonary rehabilitation, physical activity programs including exercise in water, active mind-body movement therapies, neuromuscular electrical stimulation, and personalized physical activity programs with a motivational interview can reduce exacerbations and symptoms, including dyspnea and fatigue, as well as enhance health-related QOL, along with an improvement of physical activity in patients with COPD [84,85,86,87]. COPD-related QOL, symptoms including fatigue and dyspnea, and emotional function are also improved by pulmonary rehabilitation compared to usual care [86]. Notably, high-intensity exercise training with a 30 min exercise session at 80% of baseline maximal power output improves exercise capacity with physiological adaptation to endurance training occurring if the program is completed [88,89]. Importantly, it is still unclear whether pulmonary rehabilitation contributes to improvement of physical activity via exercise capacity, because physical activity is affected by several factors, such as sociodemographic factors, lifestyle, and environmental factors, along with individual exercise capacity, as mentioned above. Therefore, pulmonary rehabilitation might contribute to improvement in physical activity, potentially increasing physical function in patients with COPD, but further data are needed.

As an extrapulmonary comorbidity, depression is important in patients with COPD [62,90], and it involves physical activity, as mentioned above. Importantly, pulmonary rehabilitation also improves anxiety and depression as examined by the Hospital Anxiety and Depression Scale [91]. These data show that pulmonary rehabilitation might improve physical activity through a psychological effect, and improvement of anxiety and depression might also improve physical activity. Unfortunately, the biological mechanisms of the beneficial effects on physical activity for patients with COPD are still unclear because of the lack of clinical studies. Therefore, prospective interventional studies of physical activity in patients with COPD, especially those focusing on biological mechanisms such as hormonal stress-responsive systems, anti-inflammatory effects, and neural effects through the brain, should be performed. 

## 4. Importance of Sedentary Behavior in Human Health

### 4.1. What Is Sedentary Behavior?

Sedentary behavior involves physically inactivity, with examples including sitting or lying down [92]. Sedentary behavior is more precisely defined as any waking behavior characterized by any energy expenditure less than or equal to 1.5 METs [16]. There is increasing evidence that sedentary behavior affects human health. The US National Health and Nutrition Examination Survey found that individuals, including children and adults, are generally sedentary for approximately 55% of their waking lives [93]. The degree to which sedentary behavior and physical activity interact in their association with health status is interesting. Biswas et al. [94] reported, in a meta-analysis, associations between physical activity and various deleterious health outcomes, and that sedentary time, independent of physical activity, is positively associated with all-cause mortality and cardiovascular disease-related mortality. Given the data, sedentary behavior, which might differ from a lack of moderate to vigorous physical activity, has qualitatively and independently harmful effects on health outcomes, human metabolism, and physical function. Katzmarzyk et al. [95] performed a prospective study to examine the relationship between sitting time and mortality in a representative sample of 17,013 Canadians ranging in age from 18 to 90 years. It was found that all-cause and cardiovascular disease-related mortalities were significantly higher if sitting time was increased, and the results remained significant after adjustment for potential confounders including physical activity level, which showed that sitting time had a harmful effect on mortality, independent of the physical activity level. Others reported that sedentary time measured by an accelerometer is associated with increases in waist circumference, triglycerides, 2 h plasma glucose, and insulin levels [96,97,98]. In addition, there is a relationship between sedentary behavior and reduced bone mass, and a 1% to 4% reduction was seen in the lumbar spine and femoral necks of healthy individuals following 12 weeks of bed rest [99]. 

Notably, sedentary behavior and physical inactivity do not have completely identical meanings, and careful interpretation of the studies mentioned above is needed, because physical inactivity constitutes an insufficient amount of moderate-to-vigorous physical activity [100], which is different from the definition of sedentary behavior, as mentioned above. In addition, a lifestyle in which time is spent performing an insufficient amount of moderate-to-vigorous physical activity does not directly involve increasing time spent being sedentary. Thus, sedentary behavior might have an impact on clinical outcomes independent of physical inactivity. According to these data, sedentary behavior is also an important independent factor along with general physical activity level, and shortening sedentary behavior time in daily life has beneficial effects on human health.

### 4.2. Factors Associated with Sedentary Behavior

Examples of sedentary behavior primarily consist of sitting and lying while watching TV, working on a computer, driving, eating, etc. There are many factors related to the duration of sedentary behavior, including sociodemographic phenotypes, such as age and sex. The amount of time spent on sedentary behavior is significantly greater with increased age in youths and adults, and it is also different between males and females in each age group among youths and adults aged 20–29, 60–69, and 70–85 years. Briefly, females spend more time sedentary than do males throughout youth and early adulthood, but this phenomenon is reversed after the age of 60 years [93]. Ethnic group is also associated with sedentary time, with Mexican Americans being less sedentary than their White counterparts in youth and adulthood [93]. Obesity, cancer, and chronic diseases including cardiovascular disease, hypertension, and diabetes mellitus, as well as psychosocial health, are also factors associated with sedentary behavior, even though whether they are the cause or the effect is unclear from the results of studies. Obesity can increase sedentary behavior because of structural and functional limitations to movement it can cause, and vice versa [101]. For children aged 7 to 11 years, TV watching and video game playing increase the risk of overweight by 17–44% and of obesity by 10–61% [102]. As for adults, Brown et al. [103] analyzed 8071 middle-aged women, and participants who spent time sitting for more than 4.5 h per day were more likely to gain more than 5 kg than those who spent time sitting less than 3 h per day [104]. Increasing TV watching time or sitting at work for 2 h per day was also associated with an increased risk of obesity in adults. In terms of the relationship between cancer and sedentary behavior, a large, prospective, cohort study of 488,720 people aged 50 to 71 years showed that longer TV and video watching time was significantly associated with increased risks of colon cancer and endometrial cancer [105,106]. In chronic disease, sedentary time is positively associated with increased risks of cardiovascular disease [107], hypertension, [108] and diabetes mellitus [109].

In terms of psychosocial health, people with the highest level of sedentary behavior showed a 31% higher risk of mental disorders compared with less sedentary individuals [110], which indicated that psychosocial health might be an important factor associated with sedentary behavior, although strong evidence, such as from a prospective intervention study, is lacking. According to these data, various factors might be associated with sedentary behavior, but biological mechanisms for reducing sedentary behavior time to obtain a beneficial effect are unclear because of the absence of interventional studies.

## 5. Clinical Impact of Sedentary Behavior in Patients with Chronic Obstructive Pulmonary Disease

### 5.1. Clinical Impact of Sedentary Behavior in Patients with COPD

Evidence and studies show that sedentary behavior is associated with clinical outcomes in patients with COPD. Sedentary behavior including sitting and lying down during the day, as measured by triaxial accelerometers, accounts for a very high percentage, approximately 64%, of patients with COPD, higher than the 46% for healthy subjects, and both time spent sitting and time spent lying down are significantly longer in patients with COPD than in healthy subjects [57]. Another retrospective, cohort study showed that sedentary behavior was independently related to mortality in patients with COPD [3]. In this study involving 101 patients with COPD, time spent on sedentary behavior, defined as <1.5 METs as measured by activity monitors, was determined, and 41 patients died during the average follow-up period of 62 months. Sedentary behavior, especially more than 8.5 h per day spent in sedentary activities at <1.5 METs, was significantly correlated with mortality in patients with COPD after adjusting for potential confounders such as sex, age, body mass index, educational level, lung function, functional exercise capacity, and moderate-to-vigorous physical activity. 

Interestingly, sedentary behavior itself might increase the risk of COPD. A large-scale, cross-sectional study of 14,073 individuals showed that those who remained sedentary for more than 7 h per day were more likely to have COPD than the control group, whose sedentary time was less than 3 h after adjustment for confounders including sex, age, country, educational level, marital status, occupation, economic status, smoking habit, physical activity, and all other chronic diseases, even though detailed physical activity levels other than sedentary time were not clarified [111]. Thus, increasing sedentary behavior time might contribute to increasing the incidence rate of COPD, along with the associated clinical outcomes.

### 5.2. Factors Possibly Associated with Sedentary Behavior in Patients with COPD

Evidence-based factors associated with sedentary behavior in patients with COPD are limited, but possible factors are considered to include COPD-related symptoms such as dyspnea and shortness of breath, reduced pulmonary function, weight loss including loss of skeletal muscle mass, decreased exercise capacity, and mental disorders such as depression. The pathophysiology of COPD involves airway narrowing as reflected by obstructive ventilatory failure on pulmonary function testing, which shows dyspnea and shortness of breath induced by dynamic pulmonary hyperinflation and oxygen desaturation [112]. Importantly, exercise can exacerbate the phenomena [113] that might contribute to avoidance of moderate or vigorous physical activity and increase sedentary behavior. Even worse, the sequence of behavior induces a negative feedback cycle leading to poor outcomes for COPD patients, because increasing sedentary time with a reduction in physical activity contributes to skeletal muscle atrophy, especially in antigravity muscles such as the erector spinae muscles and thigh muscles, causing weight loss and decreased exercise capacity [59,66,67,114]. Consequently, the symptoms of dyspnea and shortness of breath are induced at a low intensity of physical activity, and patients try to avoid movement along with progression of muscle atrophy. Thus, interventions such as bronchodilators for symptom reduction and/or pulmonary rehabilitation with behavior modification are necessary to break these negative feedback cycles and improve COPD outcomes if we are to consider therapeutic intervention to improve physical activity and decrease sedentary behavior.

Mental disorders are also associated with sedentary behavior in patients with COPD. Indeed, anxiety and depression are important comorbidities in patients with COPD, and their prevalence was 80% in patients with COPD in a US cohort [90] and 38% in Japan [115]. As mentioned above, mental disorders are known to generally contribute to increased sedentary behavior [110], meaning they are also likely associated with sedentary behavior in patients with COPD. According to these data, various factors might be associated with sedentary behavior in patients with COPD, but biological mechanisms that can inform strategies for reducing sedentary behavior time and achieve a beneficial effect still remain unclear because of the shortage of interventional studies.

## 6. Possible Interventions Focused on Physical Activity, Especially Sedentary Behavior, in Patients with Chronic Obstructive Pulmonary Disease

### 6.1. Pharmacological Intervention: Bronchodilators

Use of bronchodilators is one of the pivotal treatment modalities for patients with COPD. Several interventional studies of bronchodilators showed their potential in improving sedentary behavior. Minakata et al. [116] performed a post hoc analysis of a randomized, double-blind, active-controlled, crossover trial (VESUTO study) that evaluated the efficacy of tiotropium plus olodaterol dual therapy versus tiotropium monotherapy in Japanese patients with COPD. Sedentary behavior was measured by a three-axis accelerometer, and 182 patients were evaluated to identify the impact of the bronchodilator treatments on sedentary behavior. It was found that sedentary behavior was significantly reduced by dual therapy, with an 8.64 min grater reduction in 1.0–1.5 METs of activity per day, compared to monotherapy, along with improvement of lung function and dyspnea. 

In another study, we conducted a prospective, multicenter, randomized, open-label, parallel, interventional trial (SCOPE study) to examine the efficacy of tiotropium plus olodaterol dual therapy versus tiotropium monotherapy [117]. Importantly, the recruited patients with COPD in this study were treatment-naïve, which allowed us to effectively and specifically examine the impact of the treatments on sedentary behavior. Sedentary behavior, defined as activity of 1.0–1.5 METs measured by a triaxial accelerometer, was assessed, and 74 patients were enrolled. It was found that the duration of sedentary behavior after dual therapy tended to be reduced by more than 38.7 min per day compared to after monotherapy. Other parameters, such as changes of forced expiratory volume in 1 s and the transient dyspnea index, were more improved by dual therapy than monotherapy. We also reported that COPD patients with lower inspiratory capacity or shorter time spent on activity of more than 2.0 METs before dual therapy had a significantly greater reduction in sedentary time after dual therapy [118]. These data showed that sedentary behavior time was reduced because of improvement of decreased inspiratory capacity, which contributes to shortness of breath and dyspnea, by dual bronchodilator therapy. Recently, inhaled corticosteroid therapy was also found to be effective for patients with COPD, especially in those with frequent exacerbations with elevation of blood eosinophil levels [114,119]. We do not know the impact of inhaled corticosteroid therapy on sedentary behavior in patients with COPD, and further trials are needed to clarify its efficacy.

### 6.2. Nonpharmacological Intervention: Rehabilitation and Behavior Modification

Several interventional studies focusing on exercise rehabilitation and behavior modification showing beneficial outcomes related to sedentary behavior in patients with COPD have been reported. For example, Breyer et al. [120] reported that COPD patients were assigned to 2 h instruction by a professional Nordic walking instructor and performed a 3 month long training regimen. The training significantly reduced sitting time, and the improvement continued for 9 months after the intervention. Another trial showed that a home-based rehabilitation program and hospital-based rehabilitation program significantly reduced sedentary time and reduced acute exacerbations and emergency department visits in patients with COPD [121]. However, in a randomized, controlled trial, an intervention of ground-based walking training performed for between 30 and 45 min, two or three times a week for 8 to 10 weeks, showed no significant effect on sedentary time in patients with COPD [122]. According to these data, what constitutes effective and appropriate exercise rehabilitation to reduce sedentary time in patients with COPD is still unclear, and further interventional studies are needed.

Behavior modification might also be considered an effective intervention to improve physical activity and may be especially useful for reduction of sedentary behavior time. For example, Cruz et al. [123] reported a randomized, controlled trial in which COPD patients in the experimental group received a physical activity-focused behavioral intervention, which involved psychosocial support and education from physiotherapists, and use of a diary log to record daily steps during 3 months of pulmonary rehabilitation, with an additional 3 months of follow-up. Compared to the control group, the experimental group showed significantly reduced time of sedentary activities, along with increased moderate-to-vigorous physical activity. However, in another randomized, controlled trial performed by Cheng et al. [124], COPD patients performed a 6 week behavior change intervention that consisted of once weekly sessions for 6 weeks with a physiotherapist to reduce sedentary behavior through education, guided goal-setting, and real-time feedback. Compared to the sham intervention group, the intervention did not reduce sedentary behavior time. Considering these data, the clinical impact of exercise rehabilitation and behavior modification for sedentary behavior in patients with COPD remains controversial, and further trials are needed for clarification (Figure 1).

The heat map shows that less sitting time and more moderate-to-vigorous physical activity are related to a reduced risk of poor clinical outcomes of chronic obstructive pulmonary disease. Interventions with bronchodilator therapy and pulmonary rehabilitation with behavior modification can possibly reduce the risk, which is indicated as decreasing from the red to the green. The figure is based on the reports from the 2018 Physical Activity Guidelines Advisory Committee Scientific Report [14], with partial modification.

## 7. Conclusions

The data reviewed above show the clinical impact of physical activity and sedentary behavior in patients with COPD, along with the benefit of physical activity for healthy individuals. Sedentary behavior affects clinical outcomes of COPD independent of physical inactivity, and interventions to reduce sedentary behavior time are necessary despite the difficulty presented by the multiplicity of related factors, including lifestyle and environmental factors, along with physical disorders. To break the negative feedback cycle of worsening clinical outcomes in patients with COPD induced by increasing sedentary behavior, bronchodilator and pulmonary rehabilitation with behavior modification appear to be effective for improvement of physical activity and sedentary behavior. Further investigations, such as those with a prospective design, large population, and multilateral focus including pharmacological and nonpharmacological approaches, are needed to obtain high-level evidence.

## Figures and Tables

**Figure 1 jcm-12-01631-f001:**
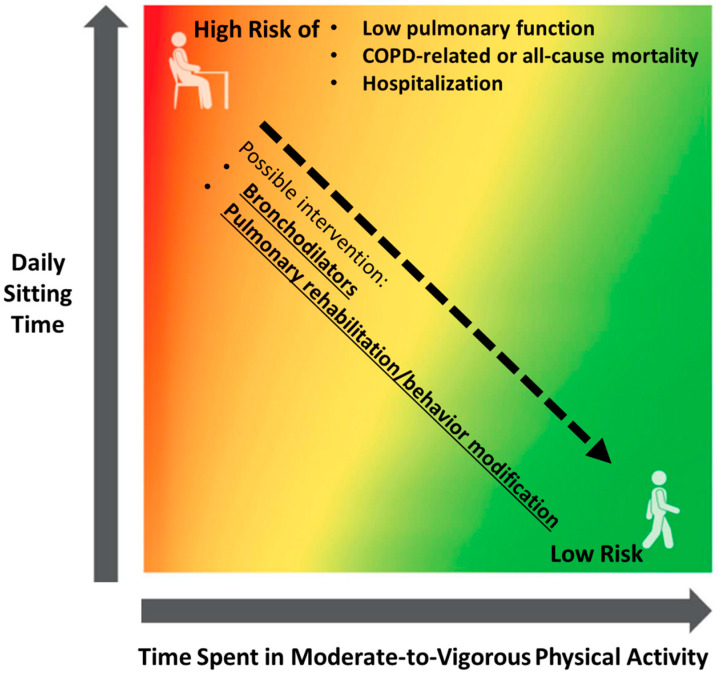
Risk of poor clinical outcomes of chronic obstructive pulmonary disease depending on physical activity and sedentary behavior, focusing on possible intervention.

**Table 1 jcm-12-01631-t001:** Examples of physical activities calculated by metabolic equivalents.

Major Headings of Physical Activity	METs
Sitting	1.5
Walking slowly	2
Standing performing light work	2–2.5
Walking at 3 mph	3.3
Walking at 4 mph	5
Walking at 4.5 mph	6.3
Jogging at 5 mph	8
Jogging at 6 mph	10
Running at 7 mph	11.5
Leisure Activities and Sports	
Arts, crafts, playing cards	1.5
Fishing, darts	2.5
Sailing, wind surfing	3
Table tennis	4
Golf	4.3
Badminton, basketball, volleyball	4.5–8
Tennis	5–8
Bicycling (10–12 mph)	6
Bicycling (14–10 mph)	10
Swimming—leisurely	6
Swimming—moderate/hard	8–11

METs: metabolic equivalents, mph: miles per hour. Data are cited from a paper by Haskell et al. [17].

## Data Availability

Not applicable.

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
