# Peer review of "Clinical Impacts of Interventions for Physical Activity and Sedentary Behavior on Patients with Chronic Obstructive Pulmonary Disease"

_jcm, 2023, doi:10.3390/jcm12041631_

Round 1
Reviewer 1 Report
The authors summarize clinical impacts and interventions of physical activity and sedentary behavior on patients with chronic obstructive pulmonary disease. In general, this review is a list of findings that have been published and is mainly lacking the critical interpretation. Of course, this is useful, but I expect more from the authors. Some of the statements are not very clear and should be more carefully formulated. Please fined some examples given below. In the same direction, I would like to see a thoughtful conclusion/discussion section where the authors’ comments more on the known factors and describe some of the possible future areas for COPD to make stronger finish for this review.
Title: Please correct the grammatical mistake in the title. The authors might consider changing the title to “Clinical impacts and interventions of physical activity and sedentary behavior on patients with chronic obstructive pulmonary disease”.
Line 26: “…………to become the third leading cause of death globally by 2030 [1].” This sentence should be rewritten. According to the WHO (https://www.who.int/news-room/fact-sheets/detail/chronic-obstructive-pulmonary-disease-(copd)) COPD was already the third leading cause of death worldwide in 2019. Furthermore, the Abstract of the reference 1 says “The United Nations (UN) has targeted a reduction of premature deaths from non-communicable diseases (NCDs) by a third by 2030.”, which means the UN has targeted to reduce the deaths from NCDs to 67% by 2030. Please refer appropriate references or rewrite this sentence.
Line 62-65: Figure or table of METs with examples of physical activities would be helpful for the readers to better grasp METs.
Line 94-96: “…the data reminds us….” Please add reference right after “the data”, because this sentence is little bit confusing to understand which fact the authors would like to refer, since “the data” is used as singular.
Line 132-133: “In addition, …… improved mood and cognition [51, 52]” I would recommend moving this sentence up to line 127 after “……… have been considered [47-50].
Line 165-170: The following sentences (1) “Exercise capacity ….…are also associated with physical activity in patients with COPD [54, 58].” and (2) “…..and factors associated with exercise capacity are also in involved in physical activity [59-61]. “ are redundant unless there is further explanation for “factors” in line 169. Please rewrite these sentences.
Line 176: Please add reference for “the Hospital Anxiety and Depression Scale”. The authors might want to refer ”The Hospital Anxiety and Depression Scale” by Zigmond and Snaith (PMID: 6880820).
Lines 244, 247, 267, and 268: “effect” can mean both positive and negative effects. It would be appreciated that if the authors describe how effect, e.g., increase, decrease, enhance, suppress, effect positively, effect, negatively, etc. , instead of using “effect” alone. The same for “different” in Lines 244 and 267 and “associated” in Line 268, to describe how different and how associated is necessary for better understanding.
Figure 1: Please clarify in the Figure if “Moderate-to Vigorous Physical Activity” indicates time of physical activity or degree of physical activity.
Grammar check: There are a lot of grammatical mistakes and run-on sentences through the manuscript but not limited lines 182-185, lines 205-2011, and lines 242-246.
Minor corrections:
Line 28: “. ……because of the results of several studies,…….” I would say “ ………. based on the results of several studies, …..”
Line 29: “….. is also important, …..” should be “……. are also important,…..”.
Line 140: “Minakata et al” Please indicate reference number.
Line 152: “Waschki et al” Please indicate reference number.
Line 152-155: “...is higher than for….” Please rewrite this sentence.
Line 181: Please define “day of the week”.
Line 185: Please make a new paragraph for biomarkers starting with this sentence “There is an increased focus on evaluation of biomarkers …….”
Lines 298-299: “Another retrospective, … in patients with COPD.” Please add reference right after this sentence.
Line 306: Please change “Nguyen” to “Nguyen et al.”
Lines 294-298: These two sentences, i.e., “First of all, …” and “In addition, …”, are redundant. Please combine them in to one sentence or rewrite these sentences.
Line 357: “Minakata et al” Please indicate reference here.
Author Response
Response to reviewers for manuscript: ID: jcm-2148460
Title: Clinical impacts of and interventions for physical activity and sedentary behavior in patients with chronic obstructive pulmonary disease
Journal: Journal of Clinical Medicine
We would like to thank the reviewers for their detailed comments and suggestions. We believe that carefully considering and responding to these comments have improved the manuscript.
We revised the manuscript which was highlighted.
Reviewer 1
The authors summarize clinical impacts and interventions of physical activity and sedentary behavior on patients with chronic obstructive pulmonary disease. In general, this review is a list of findings that have been published and is mainly lacking the critical interpretation. Of course, this is useful, but I expect more from the authors. Some of the statements are not very clear and should be more carefully formulated. Please fined some examples given below. In the same direction, I would like to see a thoughtful conclusion/discussion section where the authors’ comments more on the known factors and describe some of the possible future areas for COPD to make stronger finish for this review.
Thank you very much for the comment. We understood the necessity of our interpretation considering to the present data on this review. We carefully revised focusing on unclearly expression and reasonable conclusion. Additionally, we have double checked the grammatical mistake with native speaker and submitted the official certification of English proofreading.
Title: Please correct the grammatical mistake in the title. The authors might consider changing the title to “Clinical impacts and interventions of physical activity and sedentary behavior on patients with chronic obstructive pulmonary disease”.
We revised the title as below.
Title
Clinical impacts and interventions of physical activity and sedentary behavior on patients with chronic obstructive pulmonary disease
Line 26: “…………to become the third leading cause of death globally by 2030 [1].” This sentence should be rewritten. According to the WHO (https://www.who.int/news-room/fact-sheets/detail/chronic-obstructive-pulmonary-disease-(copd)) COPD was already the third leading cause of death worldwide in 2019. Furthermore, the Abstract of the reference 1 says “The United Nations (UN) has targeted a reduction of premature deaths from non-communicable diseases (NCDs) by a third by 2030.”, which means the UN has targeted to reduce the deaths from NCDs to 67% by 2030. Please refer appropriate references or rewrite this sentence.
Thank you very much for the pointing. We cited the website of WHO (World Health Organization, Chronic obstructive pulmonary disease (COPD) https://www.who.int/news-room/fact-sheets/detail/chronic-obstructive-pulmonary-disease-(copd)) and revised the manuscript as below.
Page 1, line 24
Chronic obstructive pulmonary disease (COPD) is a common respiratory disease, with 251 million cases worldwide, and it is a life-threatening disease that became the third leading cause of death globally in 2019 [1].
Line 62-65: Figure or table of METs with examples of physical activities would be helpful for the readers to better grasp METs.
Thank you very much for the suggestion. We made Table of METs with examples of physical activities referring to the paper below (Haskell WL, Lee IM, Pate RR, Powell KE, Blair SN, Franklin BA, Macera CA, Heath GW, Thompson PD, Bauman A. Physical activity and public health: updated recommendation for adults from the American College of Sports Medicine and the American Heart Association. Med Sci Sports Exerc. 2007 Aug;39(8):1423-34.).
Page 2, line 76
Examples of physical activities are shown in Table 1.
Table 1. Examples of physical activities calculated by metabolic equivalents.
|
Major Headings of Physical Activity |
METs |
|
Sitting |
1.5 |
|
Walking slowly |
2 |
|
Standing performing light work |
2 - 2.5 |
|
Walking at 3 mph |
3.3 |
|
Walking at 4 mph |
5 |
|
Walking at 4.5 mph |
6.3 |
|
Jogging at 5 mph |
8 |
|
Jogging at 6 mph |
10 |
|
Running at 7 mph |
11.5 |
|
Leisure Activities and Sports |
|
|
Arts, Crafts, Playing cards |
1.5 |
|
Fishing, Darts |
2.5 |
|
Sailing, Wind surfing |
3 |
|
Table tennis |
4 |
|
Golf |
4.3 |
|
Badminton, Basketball, Volleyball |
4.5 - 8 |
|
Tennis |
5 - 8 |
|
Bicycling (10 - 12 mph) |
6 |
|
Bicycling (14 - 10 mph) |
10 |
|
Swimming - leisurely |
6 |
|
Swimming - moderate/hard |
8 - 11 |
Mets: Metabolic equivalents, mph: miles per hour. Data are cited from a reference paper by Haskell et al [17].
Line 94-96: “…the data reminds us….” Please add reference right after “the data”, because this sentence is little bit confusing to understand which fact the authors would like to refer, since “the data” is used as singular.
We cited appropriate reference paper [36] and revised the manuscript considering to the confusing sentence.
Page 4, line 126
Lifestyles of individuals [36], including use of cell phones, computers, and playing video games, are also related to reductions of steps per day, and these data remind us of the necessity of ‘behavior modification’ to improve physical activity.
Line 132-133: “In addition, …… improved mood and cognition [51, 52]” I would recommend moving this sentence up to line 127 after “……… have been considered [47-50].
We transferred the sentence as reviewer suggested.
Page 5, Line 166
In addition, exercise affects the brain by enhancing growth factor expression and neural plasticity, contributing to improved mood and cognition [52, 53].
Line 165-170: The following sentences (1) “Exercise capacity ….…are also associated with physical activity in patients with COPD [54, 58].” and (2) “…..and factors associated with exercise capacity are also in involved in physical activity [59-61]. “ are redundant unless there is further explanation for “factors” in line 169. Please rewrite these sentences.
Thank you very much for the pointing of unclear expression. We carefully checked the reference paper and revised the manuscript with reference of other reviewer’s suggestion as below.
Page 5, line 182
Results on the incremental shuttle walk test and 6-minute walking distance, muscle function measured by handgrip force and quadriceps force, and symptoms such as dyspnea are also associated with physical activity in patients with COPD [57, 65]. Briefly, skeletal muscle is positively associated with exercise capacity such as assessed by 6-minute walking distance and oxygen uptake at peak exercise measured by cardiopulmonary exercise testing in patients with COPD [66, 67]. Interestingly, the cross-sectional area of skeletal muscle measured by computed tomography is positively associated with physical activity [67], showing that exercise capacity involves physical activities via skeletal muscle mass. Waschiki et al [65] reported that physical activity measured by a multisensory armband and steps per day is negatively correlated with score on the modified Medical Research Council dyspnea scale. These data show that pulmonary function, exercise tolerance involving skeletal muscles, and symptoms related to COPD, which are major pathophysiological markers and clinical features of COPD, affect physical activity.
Line 176: Please add reference for “the Hospital Anxiety and Depression Scale”. The authors might want to refer ”The Hospital Anxiety and Depression Scale” by Zigmond and Snaith (PMID: 6880820).
Thank you very much for the suggestion. We cited the reference paper as below.
Page 5, line 198
Others also found, in a prospective, multicenter study, that symptoms of depression examined by the Hospital Anxiety and Depression Scale [69] were associated with a measurable reduction in physical activity 6 months later in patients with COPD [70].
Lines 244, 247, 267, and 268: “effect” can mean both positive and negative effects. It would be appreciated that if the authors describe how effect, e.g., increase, decrease, enhance, suppress, effect positively, effect, negatively, etc. , instead of using “effect” alone. The same for “different” in Lines 244 and 267 and “associated” in Line 268, to describe how different and how associated is necessary for better understanding.
Thank you very much for the pointing. We changed the unclear expression of “effect”, “different”, “associated” or added the detailed information after the words as below.
We revised as below.
Page 7, line 277
Biswas et al [94] reported in a meta-analysis associations between physical activity and various deleterious health outcomes, and that sedentary time, independent of physical activity, is positively associated with all-cause mortality and cardiovascular dis-ease-related mortality. Given the data, sedentary behavior, which might differ from a lack of moderate to vigorous physical activity, has qualitatively and independently harmful effects on health outcomes, human metabolism, and physical function.
Page 8, line 310
The amount of time spent on sedentary behavior is significantly greater with increased age in youths and adults, and it is also different between males and females in each age group among youths and adults aged 20-29, 60-69, and 70-85 years. Briefly, females spend more time sedentary than do males throughout youth and early adulthood, but this phenomenon is reversed after the age of 60 years [93]. Ethnic group is also associated with sedentary time, with Mexican-Americans being less sedentary than their White counterparts in youth and adulthood [93].
Figure 1: Please clarify in the Figure if “Moderate-to Vigorous Physical Activity” indicates time of physical activity or degree of physical activity.
We also referred paper of Ekelund U et al (Ekelund U, et al. Lancet. 2016.) and realized that time of moderate intensity are positively associated with all-cause mortality in patients with COPD.
We revised the figure showing more clearly as described below.
Grammar check: There are a lot of grammatical mistakes and run-on sentences through the manuscript but not limited lines 182-185, lines 205-2011, and lines 242-246.
We revised the pointed lines along with whole sentences double checked by native speaker.
Minor corrections:
Line 28: “. ……because of the results of several studies,…….” I would say “ ………. based on the results of several studies, …..”
We revised the manuscript as reviewer suggested.
Page 1, line 26
Medical interventions focusing on physical activity in the stable phase are currently demanded for COPD patients based on the results of several studies, though avoiding events such as exacerbations, infections, and comorbidities are also important because they are involved in clinical outcomes, including mortality, of COPD patients [2].
Line 29: “….. is also important, …..” should be “……. are also important,…..”.
We revised the manuscript as below.
Page 1, line 26
Medical interventions focusing on physical activity in the stable phase are currently demanded for COPD patients based on the results of several studies, though avoiding events such as exacerbations, infections, and comorbidities are also important because they are involved in clinical outcomes, including mortality, of COPD patients [2].
Line 140: “Minakata et al” Please indicate reference number.
We indicated the reference number as below.
Page 5, line 182
For example, Minakata et al [56] reported that patients with COPD showed significant reduction in the duration of physical activity, including activities at each intensity level of more than 2.0 METs, 2.5 METs, 3.0 METs, and 3.5 METs, compared to healthy subjects.
Line 152: “Waschki et al” Please indicate reference number.
We indicated reference number as below.
Page 5, line 191
Importantly, Waschki et al [59] suggested, in a prospective, cohort study, that physical activity is the most important predictor of all-cause mortality.
Line 152-155: “...is higher than for….” Please rewrite this sentence.
We rewrite the unclear sentence as below.
Page 5, line 191
Importantly, Waschki et al [59] suggested, in a prospective, cohort study, that physical activity is the most important predictor of all-cause mortality. The relative risks of death with a standardized decrease in physical activity level measured by a multisensory armband and steps per day are higher than those with worsening of pulmonary function, exercise capacity assessed by 6-minute walk distance, COPD-related QOL, or symptoms such as dyspnea in patients with COPD.
Line 181: Please define “day of the week”.
We carefully checked the reference 64 (review) with the referencing paper however, we could not find information about association between physical activity levels and ‘day of the week’. Hence, we exclude that to avoid confusion.
We revised the manuscript as below.
Page 6, line 239
Lifestyle and environmental factors including alcohol consumption and smoking might also be associated with physical activity levels in patients with COPD [71].
Line 185: Please make a new paragraph for biomarkers starting with this sentence “There is an increased focus on evaluation of biomarkers …….”
We made a new paragraph with title of Possible biomarkers reflecting physical activity of patients with COPD.
Lines 298-299: “Another retrospective, … in patients with COPD.” Please add reference right after this sentence.
We followed the reviewer’s suggestion.
Line 306: Please change “Nguyen” to “Nguyen et al.”
We followed the reviewer’s suggestion.
Lines 294-298: These two sentences, i.e., “First of all, …” and “In addition, …”, are redundant. Please combine them in to one sentence or rewrite these sentences.
We combined the into one sentence as below.
Page 9, line 380
Sedentary behavior including sitting and lying down during the day, as measured by triaxial accelerometers, accounts for a very high percentage, approximately 64%, for patients with COPD, higher than the 46% for healthy subjects, and both time spent sit-ting and that spent lying down are significantly longer in patients with COPD than in healthy subjects [57].
Line 357: “Minakata et al” Please indicate reference here.
We transferred reference as below.
Page 11. Line 439
Minakata et al [116] performed a post hoc analysis of a randomized, double-blind, active controlled, crossover trial (VESUTO study) that evaluated the efficacy of tiotropium plus olodaterol dual therapy versus tiotropium monotherapy in Japanese patients with COPD.

Reviewer 2 Report
Reviewer’s comments
General comments:
1. Physical activity is an important issue and is currently hot topic in COPD and pulmonary rehabilitation.
2. Is active physical status = opposite to sedentary life? Please clarify and brief the manuscript as the writing is tedious and not straight forward even though the English language of the manuscript is linguistically acceptable.
Specific comments
Introduction
Can it be more straight forward to differentiate physical inactivity and sedentary behavior here?
Line 40: missed a period.
Line 46: "moves organs" sounds weird.
Lines 56-8: AT starts from moderate ex. intensity. what are the risk of heavy exercise? for example here.
Lines 60-: PA is usually assessed with accelerators or questionnaires (including short physical performance battery….). Please address the issues here.
Lines 67-: Vigorous intensity is usually > AT; however, is it still suitable to say aerobic? or isotonic? or dynamic? Please clarify.
Lines 68-: Any other criteria used in the literature? Such as leisure activity? Is the protocol of PA different from exercise training protocol? Is the PA protocol suitable for patients with COPD? Please commend here.
Lines 78-79: not helpful. Please delete. Lines 81-83, 86, 91 Please brief the sentences.
2.2. Does comorbidity affect sk. m. and sarcopenia or cachexia and thus affect PA? Please clarify.
Lines 87-89 and 90-94: Brief the sentences.
Line 94: Lifestyle..... (ref. please). Non-exercise? Do reductions of non-exercise activity mean increases of exercise activity? Please clarify.
Lin 98: "perceived.... such as" seems not helpful.
Lines 100 and 133: “Accordingly,” is OK?
2.3 Beneficial effects of physical activity on human health and the mechanisms
Lines 103-5: PA improves human..... Brief the sentences.
Ref 41: Does it mean that increase of PA may increase survival? or, survival is just related to their well-being (thus, more PA). Please comment on this.
Lines 114-5: Exercise effect or PA effect?
2.3. All benefits here can be achieved by exercise training (See ACSM guidelines). Is exercise training different from PA? Please clarify.
Line 122: social sports=??
Lines 127-131: Brief the sentences. Exercise training (ET) improves sk. muscle and CV. Can these effects of ET be achieved by PA?
3. Why address COPD alone but no comorbidities combined with COPD in discussion??
Minakata's ref??
Lines 142-5: shorten the sentence Lines 144-5: delete after "stage".
Shorten the sentence (lines 145-9).
The methods of measuring PA are important..... What method was used in Waschki's study? (such as a multisensory armband?) what are the symptoms (such as...)? Exercise capacity was measured with 6MWD. Please be specific.
Lines 156-7: This is a reasonable deducing but not a necessary result as PA and mortality is not necessary a cause and effect relationship. Please commend carefully. Does PR improve PA by exercise training or education? Please clarify. Missed a period.
3.2
Lines 162-3: Delete “For example….” Wrong spelling for “vital”.
Line 165: delete “Exercise capacity measured by” What is the measure of muscle function here? Please clarify. Please brief sentences for the entire manuscript. Please do not iterate.
Lines 170-1: symptoms are pathophysiological markers of COPD??
Delete lines 172-3.
Line 181: Why day of week affects PA? How this inform the readers? Please clarify.
Lines 182-4: Why say this here? help?
Line 187: 8-isoprostane??
Irisin: ref for irisin?? from Ijiri? Please clarify.
Lines 195-6: Suggest è“further studies are needed for the effects of irisin"
Lines 197-202: GDF-11 is beneficial or harmful?? not clear as it is related to inflammatory markers. (line 200) Please commend more about 8-IP, irisin, and GDF-11.
3.3. Beneficial effects of physical activity on patients with COPD and the mechanisms
Line 204: Delete “of” and replace with “in assessing all-cause........”
Lines 212-3: What was the PA program?
Line 217: Delete “As…..mechanisms”. Ref 78 is regarding the effects of pul. rehab types on ex. capacity. Please check strictly. PR should be different from PA.
Exercise training ==> improve ex capacity; but if ex capacity can translate into ==> increase PA level? motivation? environmental factors? Please clarify PA, exercise training, PR, frail, sedentary life, physical inactivity... That PR improves exercise tolerance (or capacity) is well documented.
Line 225: Delete “has ….. of” and replace with “improves”. Please brief the sentence.
Lines 226-9: PA==> improves A/D but A/D also improves PA. Please clarify.
Line 238: “specific and different”?? Please example it directly.
Lines 239-: 55% of awake time is harmful or normal? Sedentary life and PA level are opposite? One body with two sides?? Or sedentary life is different from physical inactivity level regarding mortality because sedentary life includes mild, moderate and vigorous PA levels; however, in Bisqas's study, they did not involve mild activity? Please comment why sedentary life is different from physical inactive life. Is the harmful effect on health due to sedentary life or physical inactivity level? What are the differences between SL and PI (physical inactivity)?
Line 253: Please explain why sitting time level is independent from leisure time level (or PA level?)
Down to Line 260: Exercise is good for CV, sk. m., bone, movement, psych., sleep, immunity, cancer prevention, 1' or 2', metabolism (lipid, sugar, insulin, neuropsych,,,,) Thus, it can be deduced that no exercise (i.e. sedentary life or physical inactivity) is no good for all of them. The authors may integrate sedentary life papers into the paragraphs regarding physical inactivity or not physical active. For me, the reports seem iterating, unless the authors can clearly tell the differences between sedentary life and physical inactivity.
Line 266: “older” age in youths and adults?? Not clear here. èincreased?
Line 268: Which racial/ethnic?
Line 269: Why cancer alone but not chronic diseases??
Lines 273-: What is the risk of gaining 5kg? Dose dependent? parallel description for children vs adults.
Line 275: redundant: delete "which are representative....."
Lines 281-2: redundant.
Lines 287-: Is sedentary behavior opposite to PA level?
5.
Lines 294 and 345: incidence or rate?
Lines 299-: ref??
Lines 308-: shorten the sentence. All that mentioned in COPD seems the same to non-COPD data.
Lines 314-: cause and effect?? Of course, increasing PA time is not a risk of readmission. Thus, the description here is not clear.
Lines 319-: what individuals can have <3 hr per day of sedentary life? That means they work or do activities >1.5METs >13hrs per day. Please clarify.
Lines 325-: from lung function impairment and BMI reduction to symptoms and exercise intolerance, do these progressively step-by-step develop or lump together? Hierarchically stratify? Pathophysiology of dyspnea is airflow obstruction, DH, and probably oxyhemoglobin desaturation and all of these cause dyspnea....
Not only erector spinae muscle but also the thigh muscles. Please clarify.
6.1
Line 357: ref 105?
Line 362: “bronchodilator” treatments
8.64 min??? Not clear. Please clarify.
Lines 369-70: definition of sedentary behavior is of no difference from that aforementioned. Shorten or delete the sentence here.
38.7 min per day? Not clear. Please clarify.
Lines 374-386: Shorten the paragraph.
Lines 396-8: between 30 and 45 min?? not clear.
Line 404: How was the behavior intervention conducted? Please clarify.
Figure 1: Permission from Washington DC?
Conclusions
Be more specific; it's too general comments here; but please be not lengthy when you add your comments. (for example, what are the differences in PA, exercise activity, sedentary behavior? What are the differences in improving health of healthy and diseased subjects by PA or exercise training or pul. rehab.? What further investigations are suggested?)
Author Response
Response to reviewers for manuscript: ID: jcm-2148460
Title: Clinical impacts of and interventions for physical activity and sedentary behavior in patients with chronic obstructive pulmonary disease
Journal: Journal of Clinical Medicine
We would like to thank the reviewers for their detailed comments and suggestions. We believe that carefully considering and responding to these comments have improved the manuscript.
We revised the manuscript which was highlighted.
Reviewer 2
- Physical activity is an important issue and is currently hot topic in COPD and pulmonary rehabilitation.
- Is active physical status = opposite to sedentary life? Please clarify and brief the manuscript as the writing is tedious and not straight forward even though the English language of the manuscript is linguistically acceptable.
Thank you very much for pointing. We carefully revised the manuscript following reviewer’s suggestion and correction.
Specific comments
Introduction
Can it be more straight forward to differentiate physical inactivity and sedentary behavior here?
We agree with reviewer’s suggestion to explain more about differentiation between physical inactivity and sedentary behavior. We revised the manuscript as below.
Page 1, line 31
Importantly, the effects of physical inactivity and sedentary behavior on clinical out-comes of COPD patients are different, and much data have shown that sedentary behavior negatively affects the pathophysiology of COPD, independent of physical inactivity, as described below.
Line 40: missed a period.
We added a period.
Line 46: "moves organs" sounds weird.
We revised the manuscript as below.
Page 2, line 50
Physical activity involves organs and tissues producing motion such as the skeletal muscles, heart, lungs, and others, and leads to energy metabolism via aerobic or an-aerobic pathways, depending on the individual threshold [5].
Lines 56-8: AT starts from moderate ex. intensity. what are the risk of heavy exercise? for example here.
We revised the manuscript with the examples.
Page 2, line 60
Movements inducing anaerobic metabolism are high-intensity activities that reach in excess of 90% of the maximum heart rate, increasing the risk of physical disorders such as musculoskeletal injuries along with the higher load on the body [12, 13].
Lines 60-: PA is usually assessed with accelerators or questionnaires (including short physical performance battery….). Please address the issues here.
We revised the manuscript as below
Page 2, line 66
For standardized evaluation of physical activity, several methods including question-naires and devices are used [14]. Briefly, questionnaires involve individual reporting of patients’ own physical activity behaviors, along with job categories, such as the Global physical activity questionnaire, International physical activity questionnaire, and the Short physical performance battery [14, 15]. Devices such as pedometers that count steps and accelerometers that measure truncal or limb movements are normally used [14].
Lines 67-: Vigorous intensity is usually > AT; however, is it still suitable to say aerobic? or isotonic? or dynamic? Please clarify.
We referred the expression from WHO guideline however, we agree with the reviewer’s pointing that activity of vigorous intensity level is usually more than anaerobic threshold. Hence, we exclude the word of ‘aerobic’ as below.
Page 3, line 90
For example, it is strongly recommended that children and adolescents do at least an average of 60 minutes per day of moderate- to vigorous-intensity physical activity [18-20].
Lines 68-: Any other criteria used in the literature? Such as leisure activity? Is the protocol of PA different from exercise training protocol? Is the PA protocol suitable for patients with COPD? Please commend here.
We added the information of leisure activity, recommendation of exercise training protocol.
Page 3, line 98
Muscle-strengthening activities such as exercise training at moderate or greater intensity can also be performed on 2 or more days a week, along with the physical activity protocol mentioned above [4]. In all ages, physical activity should be undertaken as part of recreation, leisure such as play, games, sports, or planned exercise, transportation, work, or household chores in daily occupational, educational, home, and community settings [4], but the appropriate intensity level of the activity should be considered depending on the physical condition, including COPD and comorbidities.
Lines 78-79: not helpful. Please delete. Lines 81-83, 86, 91 Please brief the sentences.
Thank you very much for the suggestion. We deleted the sentence.
We also revised the manuscript as below.
Page 4, line 113
Physical function, including bone mass and muscle strength, tends to decrease with increasing age, which is related to decreased physical activity. [23-25].
Page 4, line 117
Osteopenia/osteoporosis, which is characterized by low bone mass, also increases the risk of fractures due to bone fragility, leading to decreased physical activity [27, 28].
Page 4, line 122
Cross-sectional analyses involving 1,536 Germans showed that individuals who do not perform physical exercise have a 3.15 times increased risk of developing moderate to severe depression [32].
2.2. Does comorbidity affect sk. m. and sarcopenia or cachexia and thus affect PA? Please clarify.
Thank you very much for the pointing. We included the information about influence of COPD for the comorbidities such as sarcopenia, osteopenia and depression on physical activity to COPD part as indicated below.
Page 5, line 197
Notably, sarcopenia [60], osteopenia [61], and depression [62] are negatively associated with physical activity, as mentioned above, and they are major comorbidities of COPD. Thus, a decreased level of physical activity in patients with COPD might be involved in these comorbidities.
Lines 87-89 and 90-94: Brief the sentences.
We revised the manuscript as below.
Page 4, line 119
The range of motion of the ankle joint, which is important for smooth forward movement during walking, is restricted by trauma, aging, and inflammatory disease of the joint [29, 30], and importantly, limited joint mobility decreases physical activity [30].
Page 4, line 122
Mental health also contributes to physical inactivity [31]. Cross-sectional analyses in-volving 1,536 Germans showed that individuals who do not perform physical exercise have a 3.15 times increased risk of developing moderate to severe depression [32].
Line 94: Lifestyle..... (ref. please). Non-exercise? Do reductions of non-exercise activity mean increases of exercise activity? Please clarify.
We included the reference and we altered the unclear expression of ‘Non-exercise’ to steps per day as another reviewer also pointed.
We revised the manuscript as below.
Page 4, line 126
Lifestyles of individuals [36], including use of cell phones, computers, and playing video games, are also related to reductions of steps per day, and these data remind us of the necessity of ‘behavior modification’ to improve physical activity.
Lin 98: "perceived.... such as" seems not helpful.
We excluded the sentence.
Lines 100 and 133: “Accordingly,” is OK?
We revised the manuscript.
2.3 Beneficial effects of physical activity on human health and the mechanisms
Thank you very much for the suggestion. We revised the sub title as reviewer indicated.
Lines 103-5: PA improves human..... Brief the sentences.
We revised the manuscript as below.
Page 4, line 134
Improvement of physical activity produces beneficial effects on human health and quality of life (QOL) [4]. Evidence has shown that regular physical activity such as Mat Pilates improves physical function, including muscle strength, flexibility, and cardiorespiratory fitness [37].
Ref 41: Does it mean that increase of PA may increase survival? or, survival is just related to their well-being (thus, more PA). Please comment on this.
We added the manuscript as below.
Page 4, line 139
Notably, whether an intervention that increases physical activity would contribute to increasing life expectancy is not known because of the lack of clinical studies, but all-cause mortality might be reduced by improving wellbeing with increased individual physical activity.
Lines 114-5: Exercise effect or PA effect?
We revised the manuscript excluding the word of physical activity to avoid confusion.
Page 4, line 147
For hypertension, people who engage in regular exercise show decreased systolic blood pressure of approximately 12 mmHg compared to people who do not exercise regularly [41].
2.3. All benefits here can be achieved by exercise training (See ACSM guidelines). Is exercise training different from PA? Please clarify.
We added the comment of differential interpretation between exercise training and physical activity level.
Page 5, line 158
Notably, exercise as mentioned above is defined as planned, structured, and repetitive physical activity, not exactly identical to the broad meaning of physical activity [47]. However, exercise training has the capacity to improve physical activity with modification of behavior [47].
Line 122: social sports=??
We added the information of social sports.
Page 5, line 155
Similarly, a walking program in addition to social sports, measured using a 10-item modified version of the social support for exercise scale [45], was significantly associated with greater positive mood in women [46].
Lines 127-131: Brief the sentences. Exercise training (ET) improves sk. muscle and CV. Can these effects of ET be achieved by PA?
Thank you very much for the pointing. We revised the manuscript as below.
Page 5, line 169
Briefly, cancers, hypertension, and type 2 diabetes mellitus are associated with systemic markers of inflammation such as tumor necrosis factor-, interleukin (IL)-1, IL-6, IL-8, and C-reactive protein, and exercise intervention reduces them [48]. As men-tioned above, exercise intervention increases physical activity along with improvements of skeletal muscle function and cardiopulmonary function, and the improvement might attenuate systemic inflammation [54, 55].
- Why address COPD alone but no comorbidities combined with COPD in discussion??
We included information of association between comorbidities of COPD and physical activity in patients with COPD.
Page 5, line 197
Notably, sarcopenia [60], osteopenia [61], and depression [62] are negatively associated with physical activity, as mentioned above, and they are major comorbidities of COPD. Thus, a decreased level of physical activity in patients with COPD might be involved in these comorbidities.
Minakata's ref??
We included the missing reference number as below.
Lines 142-5: shorten the sentence Lines 144-5: delete after "stage".
We revised the manuscript as below.
Page 5, line 185
In addition, levels of physical activity were further reduced with progression of the Global Initiative for Chronic Obstructive Lung Disease (GOLD) stage [56].
Shorten the sentence (lines 145-9).
We revised the manuscript as below.
Page 5, line 186
Similarly, walking and standing times are shorter, and time spent sitting and lying down is longer in patients with COPD than in healthy subjects [57].
The methods of measuring PA are important..... What method was used in Waschki's study? (such as a multisensory armband?) what are the symptoms (such as...)? Exercise capacity was measured with 6MWD. Please be specific.
Thank you very much for the pointing. We revised the manuscript as below.
Page 5, line 191
Importantly, Waschki et al [59] suggested, in a prospective, cohort study, that physical activity is the most important predictor of all-cause mortality. The relative risks of death with a standardized decrease in physical activity level measured by a multisensory armband and steps per day are higher than those with worsening of pulmonary function, exercise capacity assessed by 6-minute walk distance, COPD-related QOL, or symptoms such as dyspnea in patients with COPD.
Lines 156-7: This is a reasonable deducing but not a necessary result as PA and mortality is not necessary a cause and effect relationship. Please commend carefully. Does PR improve PA by exercise training or education? Please clarify. Missed a period.
We carefully revised the manuscript according to the reviewer’s comment.
Page 6, line 204
Again, it is unknown whether interventions that increase physical activity in patients with COPD, such as pulmonary rehabilitation, would directly contribute to improvement in the clinical outcomes of COPD, such as mortality, QOL, symptoms, and exacerbations, because of a shortage of data.
3.2
Lines 162-3: Delete “For example….” Wrong spelling for “vital”.
We revised the manuscript.
Line 165: delete “Exercise capacity measured by” What is the measure of muscle function here? Please clarify. Please brief sentences for the entire manuscript. Please do not iterate.
Thank you very much for pointing. We revised the manuscript as both reviewers suggested.
Page 6, line 217
Results on the incremental shuttle walk test and 6-minute walking distance, muscle function measured by handgrip force and quadriceps force, and symptoms such as dyspnea are also associated with physical activity in patients with COPD [57, 65]. Briefly, skeletal muscle is positively associated with exercise capacity such as assessed by 6-minute walking distance and oxygen uptake at peak exercise measured by cardiopulmonary exercise testing in patients with COPD [66, 67]. Interestingly, the cross-sectional area of skeletal muscle measured by computed tomography is positively associated with physical activity [67], showing that exercise capacity involves physical activities via skeletal muscle mass.
Lines 170-1: symptoms are pathophysiological markers of COPD??
We revised the manuscript as below.
Page 6, line 228
These data show that pulmonary function, exercise tolerance involving skeletal muscles, and symptoms related to COPD, which are major pathophysiological markers and clinical features of COPD, affect physical activity.
Delete lines 172-3.
We deleted the sentence as reviewer suggested.
Line 181: Why day of week affects PA? How this inform the readers? Please clarify.
We carefully checked the reference 64 (review) with the referencing paper however, we could not find information about association between physical activity levels and ‘day of the week’. Hence, we exclude that to avoid confusion.
We revised the manuscript as below.
Page 6, line 239
Lifestyle and environmental factors including alcohol consumption and smoking might also be associated with physical activity levels in patients with COPD [71].
Lines 182-4: Why say this here? help?
We excluded the sentence.
Line 187: 8-isoprostane??
We revised the manuscript as below.
Page 6, line 244
For example, the total cholesterol level in blood, which reflects cardiac function and nutritional status, 8-isoprostane in exhaled breath condensate (EBC), which is an airway oxidative stress marker, and IL-6 in EBC, which reflects systemic inflammation, are negatively correlated with physical activity level [68, 72].
Irisin: ref for irisin?? from Ijiri? Please clarify.
We added the information with references as below.
Page 6, line 248
Myokines, especially irisin, which has been discovered as a hormone secreted from skeletal myocytes at the start of exercise training [73, 74] is considered a valuable biomarker reflecting physical activity in COPD.
Lines 195-6: Suggest è“further studies are needed for the effects of irisin"
We revised the manuscript as below.
Page 7, line 254
suggesting that irisin might be a useful candidate biomarker reflecting physical activity in patients with COPD.
Lines 197-202: GDF-11 is beneficial or harmful?? not clear as it is related to inflammatory markers. (line 200) Please commend more about 8-IP, irisin, and GDF-11.
We revised the manuscript as below.
Page 7, line 257
Others have reported that growth differentiation factor 11 (GDF-11) in plasma, which is expressed in skeletal muscle and is linked to rejuvenating effects such as muscle regeneration, was positively correlated with physical activity level. GDF-11 is increased with improvements of lung function, quadriceps strength, and exercise capacity, and decreased inflammatory markers in patients with COPD [77]. According to these data, irisin and GDF-11 are beneficial, and 8-isoprostane has a harmful effect for COPD patients with respect to physical activity.
3.3. Beneficial effects of physical activity on patients with COPD and the mechanisms
We revised the subtitle as reviewer suggested.
Line 204: Delete “of” and replace with “in assessing all-cause........”
We revised the manuscript as below.
Page 7, line 266
Previous epidemiological data have shown that increased physical activity is associ-ated with the possible improvement of all-cause mortality or COPD-related mortality [58, 59].
Lines 212-3: What was the PA program?
We revised the manuscript as below.
Page 7, line 274
There are several lines of evidence showing that pulmonary rehabilitation, physical activity programs including exercise in water, active mind-body movement therapies, neuromuscular electrical stimulation, and personalized physical activity programs with a motivational interview can reduce exacerbations, symptoms, including dyspnea and fatigue, and enhance health-related QOL along with improvement of physical ac-tivity in patients with COPD [84-87].
Line 217: Delete “As…..mechanisms”. Ref 78 is regarding the effects of pul. rehab types on ex. capacity. Please check strictly. PR should be different from PA.
We understood differences between pulmonary rehabilitation and physical activity. Hence, we deleted the sentence with ref 78
Exercise training ==> improve ex capacity; but if ex capacity can translate into ==> increase PA level? motivation? environmental factors? Please clarify PA, exercise training, PR, frail, sedentary life, physical inactivity... That PR improves exercise tolerance (or capacity) is well documented.
We added the information in the manuscript as below.
Page 7, line 284
Importantly, it is still unclear whether pulmonary rehabilitation contributes to improvement of physical activity via exercise capacity, because physical activity is af-fected by several factors, such as sociodemographic factors, lifestyle, and environmental factors, along with individual exercise capacity, as mentioned above. Therefore, pulmonary rehabilitation might contribute to improvement in physical activity, potentially increasing physical function in patients with COPD, but further data are needed.
Line 225: Delete “has ….. of” and replace with “improves”. Please brief the sentence.
We revised the manuscript as below.
Page 7, line 292
Importantly, pulmonary rehabilitation also improves anxiety and depression as examined by the Hospital Anxiety and Depression Scale [91].
Lines 226-9: PA==> improves A/D but A/D also improves PA. Please clarify.
We revised the manuscript as below.
Page 7, line 294
These data show that pulmonary rehabilitation might improve physical activity through a psychological effect, and improvement of anxiety and depression might also improve physical activity.
Line 238: “specific and different”?? Please example it directly.
We excluded the unclear expressions of ‘specific and different’ because several examples were shown below the sentence.
We revised the manuscript as below.
Page 8, line 307
There is increasing evidence that sedentary behavior affects human health.
Lines 239-: 55% of awake time is harmful or normal? Sedentary life and PA level are opposite? One body with two sides?? Or sedentary life is different from physical inactivity level regarding mortality because sedentary life includes mild, moderate and vigorous PA levels; however, in Bisqas's study, they did not involve mild activity? Please comment why sedentary life is different from physical inactive life. Is the harmful effect on health due to sedentary life or physical inactivity level? What are the differences between SL and PI (physical inactivity)?
55% is normal rate of sedentary behavior. We revised the manuscript and also addressed about differences between sedentary life and physical inactivity even though it might be difficult to clearly identify the differences.
Page 8, line 307
There is increasing evidence that sedentary behavior affects human health. The U.S. National Health and Nutrition Examination Survey found that individuals, including children and adults, are generally sedentary for approximately 55% of their waking lives [93].
Page 8, line 330
Notably, sedentary behavior and physical inactivity do not have completely identical meanings, and careful interpretation of the studies mentioned above is needed, be-cause physical inactivity constitutes an insufficient amount of moderate-to-vigorous physical activity [100], which is different from the definition of sedentary behavior, as mentioned above. In addition, a lifestyle in which time is spent performing an insufficient amount of moderate-to-vigorous physical activity does not directly involve in-creasing time spent being sedentary. Thus, sedentary behavior might have an impact on clinical outcomes independent of physical inactivity.
Line 253: Please explain why sitting time level is independent from leisure time level (or PA level?)
They indicated that all-cause mortality and cardiovascular disease-related mortality are significantly higher if the sitting time is increased and those results are still significant after adjustment by cofounders including physical activity level on multivariate analysis.
We revised the manuscript as below.
Page 8, line 320
It was found that all-cause and cardiovascular disease-related mortalities were significantly higher if sitting time was increased, and the results remained significant after adjustment for potential confounders including physical activity level, which showed that sitting time had a harmful effect on mortality, independent of the physical activity level.
Down to Line 260: Exercise is good for CV, sk. m., bone, movement, psych., sleep, immunity, cancer prevention, 1' or 2', metabolism (lipid, sugar, insulin, neuropsych,,,,) Thus, it can be deduced that no exercise (i.e. sedentary life or physical inactivity) is no good for all of them. The authors may integrate sedentary life papers into the paragraphs regarding physical inactivity or not physical active. For me, the reports seem iterating, unless the authors can clearly tell the differences between sedentary life and physical inactivity.
We revised the manuscript focusing on deference between sedentary behavior and physical inactivity as above which we believe the paragraph described more focus on sedentary behavior independent from physical inactivity.
Page 8, line 330
Notably, sedentary behavior and physical inactivity do not have completely identical meanings, and careful interpretation of the studies mentioned above is needed, be-cause physical inactivity constitutes an insufficient amount of moderate-to-vigorous physical activity [100], which is different from the definition of sedentary behavior, as mentioned above. In addition, a lifestyle in which time is spent performing an insufficient amount of moderate-to-vigorous physical activity does not directly involve in-creasing time spent being sedentary. Thus, sedentary behavior might have an impact on clinical outcomes independent of physical inactivity.
Line 266: “older” age in youths and adults?? Not clear here. èincreased?
We revised the manuscript as below.
Page 9, line 345
The amount of time spent on sedentary behavior is significantly greater with increased age in youths and adults, and it is also different between males and females in each age group among youths and adults aged 20-29, 60-69, and 70-85 years.
Line 268: Which racial/ethnic?
We revised the manuscript as below
Page 9, line 356
Ethnic group is also associated with sedentary time, with Mexican-Americans being less sedentary than their White counterparts in youth and adulthood [93].
Line 269: Why cancer alone but not chronic diseases??
We addressed the information of chronic diseases including cardiovascular disease, hypertension, diabetes and revised the manuscript as below.
Page 9, line 351
Obesity, cancer, and chronic diseases including cardiovascular disease, hypertension, and diabetes mellitus, as well as psychosocial health are also factors associated with sedentary behavior, even though whether they are the cause or the effect is unclear from the results of studies.
Page 9, line 365
In chronic disease, sedentary time is positively associated with increased risks of cardiovascular disease [107], hypertension, [108] and diabetes mellitus [109].
Lines 273-: What is the risk of gaining 5kg? Dose dependent? parallel description for children vs adult.
We revised the manuscript and the reference data does not show dose dependent effect. We also described parallel for adult and children
Page 9, line 356
For children aged 7 to 11 years, TV watching and video game playing increase the risk of overweight by 17-44% and of obesity by 10-61% [102]. As for adults, Brown et al [103] analyzed 8071 middle-aged women, and participants who spent time sitting for more than 4.5 hours per day were more likely to gain more than 5 kg than those who spent time sitting less than 3 hours per day [104]. Increasing TV watching time or sit-ting at work for 2 hours per day was also associated with an increased risk of obesity in adults.
Line 275: redundant: delete "which are representative....."
We deleted the sentence.
Lines 281-2: redundant.
We excluded the sentence.
Lines 287-: Is sedentary behavior opposite to PA level?
We mentioned above by the revision.
Lines 294 and 345: incidence or rate?
We revised the manuscript as below.
Page 9, line 380
Sedentary behavior including sitting and lying down during the day, as measured by triaxial accelerometers, accounts for a very high percentage, approximately 64%, for patients with COPD, higher than the 46% for healthy subjects, and both time spent sit-ting and that spent lying down are significantly longer in patients with COPD than in healthy subjects [57].
Page 10, line 424
Mental disorders are also associated with sedentary behavior in patients with COPD. Indeed, anxiety and depression are important comorbidities in patients with COPD, and their prevalence was 80% in patients with COPD in a US cohort [90] and 38% in Japan [115].
Lines 299-: ref??
We added the reference and revised the manuscript as below.
Page 9, line 384
Another retrospective, cohort study showed that sedentary behavior was independently related to mortality in patients with COPD [3]. In this study involving 101 patients with COPD, time spent on sedentary behavior, defined as less than 1.5 METs as measured by activity monitors, was determined, and 41 patients died during the average follow-up period of 62 months. Sedentary behavior, and especially more than 8.5 hours per day spent in sedentary activities at less than 1.5 METs, was significantly correlated with mortality in patients with COPD after adjusting for potential confounders such as sex, age, body mass index, educational level, lung function, functional exercise capacity, and moderate-to-vigorous physical activity.
Lines 308-: shorten the sentence. All that mentioned in COPD seems the same to non-COPD data.
We realized that the data is focused on physical inactivity but not sedentary behavior and we decided that the reference is not suitable for the paragraph. Hence, we excluded the whole sentence related to the reference.
Lines 314-: cause and effect?? Of course, increasing PA time is not a risk of readmission. Thus, the description here is not clear.
We excluded the sentence as we mentioned above.
Lines 319-: what individuals can have <3 hr per day of sedentary life? That means they work or do activities >1.5METs >13hrs per day. Please clarify.
In that reference paper, control group whose sedentary time was less than 3 hours is not clarified how they spent other than the 3 hours. Hence, we mentioned that in the manuscript.
Page 10, line 393
Interestingly, sedentary behavior itself might increase the risk of COPD. A large-scale, cross-sectional study of 14,073 individuals showed that those who remained sedentary for more than 7 hours per day were more likely to have COPD than the control group whose sedentary time was less than 3 hours after adjustment for confounders including sex, age, country, educational level, marital status, occupation, economic status, smoking habit, physical activity, and all other chronic diseases, even though detailed physical activity levels other than sedentary time were not clarified [111].
Lines 325-: from lung function impairment and BMI reduction to symptoms and exercise intolerance, do these progressively step-by-step develop or lump together? Hierarchically stratify? Pathophysiology of dyspnea is airflow obstruction, DH, and probably oxyhemoglobin desaturation and all of these cause dyspnea....
We do agree with the reviewer’s idea that factors including lung function, BMI, exercise tolerance, desaturation and respiratory symptoms are related together. Additionally, we consider that decreasing of pulmonary function (as definition of COPD) might induce exacerbation of dyspnea, desaturation à decreasing of physical activity à body weight reduction by muscle atrophy and increased energy consumption (for breathing and systemic inflammation) à decreasing of exercise intolerance. However, those net interactions are complicated and we do not have enough data to show accurate strategy in terms of sedentary behavior. Hence, we did not touch the hierarchical pathophysiology.
Not only erector spinae muscle but also the thigh muscles. Please clarify.
We revised the manuscript as below.
Page 10, line 413
Even worse, the sequence of behavior induces a negative feedback cycle leading to poor outcomes for COPD patients, because increasing sedentary time with reduction of physical activity contributes to skeletal muscle atrophy, especially in anti-gravity muscles such as the erector spinae muscles and thigh muscles, causing weight loss and decreased exercise capacity [59, 66, 67, 114].
6.1
Line 357: ref 105?
We transferred the reference number right after ‘Minakata et al’ as shown below.
Page 11, line 439
Minakata et al [116] performed a post hoc analysis of a randomized, double-blind, active-controlled, crossover trial (VESUTO study) that evaluated the efficacy of tiotropium plus olodaterol dual therapy versus tiotropium monotherapy in Japanese patients with COPD.
Line 362: “bronchodilator” treatments
We revised the manuscript as below.
Page 11, line 442
Sedentary behavior was measured by a three-axis accelerometer, and 182 patients were evaluated to identify the impact of the bronchodilator treatments on sedentary behavior. It was found that sedentary behavior was significantly reduced by dual therapy, with an 8.64-minute grater reduction in 1.0-1.5 METs activity per day, compared to monotherapy, along with improvement of lung function and dyspnea.
8.64 min??? Not clear. Please clarify.
We revised the manuscript as below.
Page 11, line 442
Sedentary behavior was measured by a three-axis accelerometer, and 182 patients were evaluated to identify the impact of the bronchodilator treatments on sedentary behavior. It was found that sedentary behavior was significantly reduced by dual therapy, with an 8.64-minute grater reduction in 1.0-1.5 METs activity per day, compared to monotherapy, along with improvement of lung function and dyspnea.
Lines 369-70: definition of sedentary behavior is of no difference from that aforementioned. Shorten or delete the sentence here.
We deleted as below.
Page 11, line 442
Sedentary behavior was measured by a three-axis accelerometer, and 182 patients were evaluated to identify the impact of the bronchodilator treatments on sedentary behavior. It was found that sedentary behavior was significantly reduced by dual therapy, with an 8.64-minute grater reduction in 1.0-1.5 METs activity per day, compared to monotherapy, along with improvement of lung function and dyspnea.
38.7 min per day? Not clear. Please clarify.
It was found that the duration of sedentary behavior after dual therapy tended to be reduced by more than 38.7 minutes per day compared to after monotherapy.
Lines 374-386: Shorten the paragraph.
We shorten the paragraph as below.
Page 11, line 458
We also reported that COPD patients with lower inspiratory capacity or shorter time spent on activity of more than 2.0 METs before dual therapy had significantly greater reduction of sedentary time after dual therapy [118]. These data showed that sedentary behavior time was reduced because of improvement of decreased inspiratory capacity, which contributes to shortness of breath and dyspnea, by dual bronchodilator therapy.
Lines 396-8: between 30 and 45 min?? not clear.
We revised the manuscript as below.
Page 11. Line 477
However, in a randomized, controlled trial, an intervention of ground-based walking training performed for between 30 and 45 minutes, two or three times a week for 8 to 10 weeks, showed no significant effect on sedentary time in patients with COPD [122].
Line 404: How was the behavior intervention conducted? Please clarify.
We revised the manuscript as below.
Page 11, line 485
Cruz et al [123] reported a randomized, controlled trial in which COPD patients in the experimental group received a physical activity-focused behavioral intervention, which involved psychosocial support and education from physiotherapists, and use of a diary log to record daily steps during 3 months of pulmonary rehabilitation, with an additional 3 months of follow-up.
Figure 1: Permission from Washington DC?
We got permission from Director of U.S. Department of Health and Human Services (HHS) and she indicated that only citation is needed in manuscript because the materials in the 2018 Physical Activity Advisory Committee Scientific Report are in the public domain.
Conclusions
Be more specific; it's too general comments here; but please be not lengthy when you add your comments. (for example, what are the differences in PA, exercise activity, sedentary behavior? What are the differences in improving health of healthy and diseased subjects by PA or exercise training or pul. rehab.? What further investigations are suggested?)
We revised the conclusion as below.
Page 12, line 509
- Conclusions
The data reviewed above show the clinical impact of physical activity and sedentary behavior in patients with COPD, along with the benefit of physical activity for healthy individuals. Sedentary behavior affects clinical outcomes of COPD independent of physical inactivity, and interventions to reduce sedentary behavior time are necessary despite the difficulty presented by the multiplicity of related factors, including lifestyle and environmental factors, along with physical disorders. To break the negative feedback cycle of worsening clinical outcomes in patients with COPD induced by increasing sedentary behavior, bronchodilator and pulmonary rehabilitation with behavior modification appear to be effective for improvement of physical activity and sedentary behavior. Further investigations, such as those with a prospective design, large population, and multilateral focus including pharmacological and non-pharmacological approaches, are needed to obtain high-level evidence.

Round 2
Reviewer 1 Report
This review article has improved a lot. The authors have addressed all of my concerns with the original manuscript. The current version of review is well summarized and is suitable for publication.
Author Response
Response to reviewer 1 (round 2)
manuscript: ID: jcm-2148460
Journal: Journal of Clinical Medicine
Title: Clinical impacts of and interventions for physical activity and sedentary behavior in patients with chronic obstructive pulmonary disease
We would like to thank the reviewer’s comments.
